# Evaluating a Stochastic Parametrization of the Coupling in a Fast-Slow System using the Wasserstein Distance

Gabriele Vissio[1,2] and Valerio Lucarini[2,3,4]

[1]International Max Planck Research School on Earth System Modelling, Hamburg, Germany
[2]CEN, Meteorological Institute, University of Hamburg, Hamburg, Germany
[3]Department of Mathematics and Statistics, University of Reading, Reading, UK
[4]Walker Institute for Climate System Research, University of Reading, Reading, UK

**Correspondence:** Gabriele Vissio (gabriele.vissio@mpimet.mpg.de)

**Abstract.** Constructing accurate, flexible, and efficient parametrizations is one of the great challenges in the numerical modelling of geophysical fluids. We consider here the simple yet paradigmatic case of a Lorenz 84 model forced by a Lorenz 63 model and derive a parametrization using a recently developed statistical mechanical methodology based on the Ruelle response theory. We derive an expression for the deterministic and the stochastic component of the parametrization and we show that the approach allows for dealing seamlessly with the case of the Lorenz 63 being a fast as well as a slow forcing compared to the characteristic time scales of the Lorenz 84 model. We test our results using both standard metrics based on the moments of the variables of interest as well as Wasserstein distance between the projected measure of the original system on the Lorenz 84 model variables and the measure of the parametrized one. By testing our methods on reduced phase spaces obtained by projection, we find support to the idea that comparisons based on the Wasserstein distance might be of relevance in many applications despite the curse of dimensionality.

## 1 Introduction

The climate is a forced and dissipative system featuring variability on a large range of spatial and temporal scales, as a result of many complex and coupled dynamical processes inside it (Peixoto and Oort, 1992; Lucarini et al., 2014a; Ghil, 2015). Numerical models are able to resolve explicitly only a relatively small range of such scales. In particular, it is crucial to derive efficient and accurate ways to surrogate the effect of dynamical processes occurring on the small spatial and temporal scales that are not explicitly resolved (e.g. because of excessive computational or storage costs) by the model. The operation of constructing so-called parametrizations is key to the development of geophysical fluid dynamical models and stimulates the investigation of the fundamental laws defining the multiscale properties of the coupled atmosphere-ocean dynamics (Uboldi and Trevisan, 2015; Vannitsem and Lucarini, 2016). Traditionally, the development of parametrizations boiled down to deriving deterministic empirical laws able to describe the effect of the small scale dynamical processes. More recently, it has become apparent the need to include stochastic terms able to provide a theoretically more coherent representation of such effects and, at practical level, an improved skill (Palmer and Williams, 2008; Franzke et al., 2015; Berner et al., 2017).

A first way to derive or at least justify the need for stochastic parametrizations comes from homogenization theory (Pavliotis and Stuart, 2008), which leads to constructing an approximate representation of the impact of the fast scales on the slow variables as the sum of two terms, a mean field term and a white noise term. Such an approach suffers from the fact that one has to take the rather nonphysical hypothesis that an infinite time scale separation exists between the fast and the slow scale. As the climate is a multiscale system, such a methodology is a bit problematic to adopt. Yet, this point of view has been crucial in the development of methods aimed at deriving reduced order models for system of geophysical interest (see, e.g., Majda et al. (1999, 2001, 2003); Franzke et al. (2005)).

Mori et al. (1974) and Zwanzig (1960, 1961) analyzed, in the context of statistical mechanics, the related problem of studying how one can project out the effect of a group of variables, with the goal of constructing effective evolution equations for a subset of variables of interest. They reformulated the dynamics of such variables expressing them as a sum of three terms, a deterministic term, a stochastic forcing and a memory term. The memory term defines a non-markovian contribution where the past states of the variables of interest enter the evolution equation. In the limit of infinite time scale separation such last term tends to zero, whilst the random forcing approaches the form of a (in general, multiplicative) white noise.

The triad of terms - deterministic, stochastic and non-markovian - was also found by Wouters and Lucarini (2012), who proposed a method (we refer to it in what follows as WL parametrization) for constructing parametrizations based on the Ruelle response theory (Ruelle, 1998, 2009). They interpreted the coupling between the variables of interest and those one wants to parametrize as a weak perturbation of the otherwise unperturbed dynamics of the two groups of variables. A useful feature of this approach is that it can be applied on a wide variety systems that do not feature a clear-cut separation of scales. The parametrizations obtained along these lines match the result of the perturbative expansion of the projection operator introduced by Mori and Zwanzig for describing the effective dynamics of the variables of interest (Wouters and Lucarini, 2013, 2016). Another quality of the WL paramerization is that it is not tailored to optimise the representation of the statistics of some specific statistical property, but rather approximates coherently well all observables of the system of interest. This method has already been successfully tested in simple to intermediate complexity multiscale models by Wouters et al. (2016); Demaeyer and Vannitsem (2017); Vissio and Lucarini (2018).

Conceptually similar results have been found through bottom up, data driven approaches, by Kravtsov et al. (2005); Chekroun et al. (2015a, b); Kondrashov et al. (2015). Specifically, Kravtsov et al. (2005) constructed effective models from climatic time series through an extension to the non linear case of the multilevel linear regressive method, while Kondrashov et al. (2015) showed how non-markovian data-driven parametrizations emerge naturally when we consider partial observations from a large-dimensional system.

Even when a parametrization is efficient enough to represent unresolved phenomena with the desired precision, problems arise when it comes to dealing with scale adaptivity. Re-tuning the parametrization to a new set of parameters of the model usually means running again long simulations, adding further computational costs. For this reason the development of a scale adaptive parametrization is considered to be a central task in geosciences (Arakawa et al., 2011; Park, 2014; Sakradzija et al., 2016). In a previous paper, the authors demonstrated the scale adaptivity of the WL approach by testing it in a mildly modified version of the Lorenz 96 model (Lorenz, 1996). A further degree of flexibility of this approach has been explored in another

recent publication (Lucarini and Wouters, 2017), who provided explicit formulas for modifying the parametrization when the parameters controlling the dynamics of the full system are altered.

In this paper, we wish to apply the WL parametrization to a simple dynamical system introduced by Bódai et al. (2011) and constructed by coupling the Lorenz 84 (Lorenz, 1984) model with the Lorenz 63 (Lorenz, 1963) model. In what follows, we want to parametrize the dynamical effect of the variables corresponding to the Lorenz 63 system on the variables corresponding to the Lorenz 84 system. We analyse two different scenarios, where the Lorenz 63 model acts as a fast and as a slow forcing, respectively, taking into account that the WL parametrization is adaptive and able to seamlessly treat both of them. Compared to what studied in Vissio and Lucarini (2018), the models investigated here have a less rich dynamics, as they are not spatially extended, and their coupling is simpler, since it is only one-way. Nonetheless, we propose a significant advance with respect to our previous work in terms of methodology for evaluating the performance of the parametrization. We wish to extend what studied in Vissio and Lucarini (2018) by focusing on doing a systematic comparison of the properties of the projected measure of the original coupled system on the subspace spanned by the variables of the Lorenz 84 model with the actual measure of the parametrized model. In particular, we will study the Wasserstein distance (Villani, 2009) between the coarse-grained estimates of the two 3-dimensional invariant measures. Additionally, we will look at the Wasserstein distance of the measures obtained by projecting on two of the three variables of interest, which allows for a comprehensive evaluation of how different the one-time statistical properties of the two systems are. The Wasserstein distance has been proposed by Ghil (2015) as a tool for studying the climate variability and response to forcings, and applied by Robin et al. (2017) in a simplified setting.

In Section 2 we describe thoroughly the individual models and the full coupled model, while in Section 3 we briefly review Wouters-Lucarini's parametrization and its application to the Lorenz 84-Lorenz 63 coupled model. Section 4 is dedicated to discussing the Wasserstein distance and in particular a) whether it is efficient in summarizing the quality of the parametrization, b) how sensitive our analysis is to the coarse-graining of the phase space, and c) whether useful conclusions can be drawn by looking at the problem in a projected space of two variables only. Section 5 provides the main results of our analysis. In the last Section we draw our conclusions and propose future investigations.

## 2 Models

### 2.1 Lorenz 84

The Lorenz 84 model (Lorenz, 1984) provides an extremely simplified representation of the large scale atmospheric circulation:

$$\frac{dX}{dt} = -Y^2 - Z^2 - aX + aF_0, \tag{1}$$

$$\frac{dY}{dt} = XY - bXZ - Y + G, \tag{2}$$

$$\frac{dZ}{dt} = XZ + bXY - Z. \tag{3}$$

where the variable $X$ describes the intensity of the westerlies, while the variables $Y$ and $Z$ correspond to the two phases of the planetary waves responsible for the meridional heat transport. Thus, Eq.(1) describes the evolution of the westerlies, subject to

the external forcing $F_0$, dampened both by the linear term $-aX$ and by nonlinear interaction with the eddies $-Y^2$ and $-Z^2$. This interaction amplifies the eddies through the terms $XY$ and $XZ$ in Eqs.(2)-(3). Furthermore, the eddies are affected by the westerlies through the terms $-bXZ$ and $bXY$. The constant $b$ regulates the relative time scale between diplacements and amplifications. In Eqs.(2)-(3) we can, as in Eq.(1), see a linear dissipation, whilst the symmetry between the two equations is broken by the external forcing $G$.

## 2.2 Lorenz 63

The Lorenz 63 model is probably the most iconic chaotic dynamical system (Saltzman, 1962; Lorenz, 1963; Ott, 1993) and was developed through a severe truncation of the partial differential equations describing the Rayleigh-Benard problem (see e.g. Hilborn (2000) for a complete, yet simple, derivation of the model) and describe the evolution of three modes corresponding to large scale motions and temperature modulations in the Rayleigh-Bénard problem. The three equations are:

$$\frac{d\widetilde{x}}{dt} = s(\widetilde{y} - \widetilde{x}), \tag{4}$$

$$\frac{d\widetilde{y}}{dt} = \rho\widetilde{x} - \widetilde{y} - \widetilde{x}\widetilde{z}, \tag{5}$$

$$\frac{d\widetilde{z}}{dt} = -\beta\widetilde{z} + \widetilde{x}\widetilde{y}, \tag{6}$$

where $\widetilde{x}$, $\widetilde{y}$ and $\widetilde{z}$ are proportional, respectively, to the intensity of the convective motions, to the difference between temperatures of upward and downward fluid flows and to the difference of the temperature in the center of a convective cell with respect to a linear profile (since Eqs.(5)-(6) derive from thermal diffusion equation). The constants $s$, $\rho$ and $\beta$ are constants which depend on kinematic viscosity, thermal conductivity, depth of the fluid, gravity acceleration, thermal expansion coefficient; specifically, $s$ is also known as the *Prandtl Number*.

## 2.3 Coupled model

The full model used in this paper, proposed by Bódai et al. (2011), is constructed by coupling the two low-order models introduced before as follows. The Lorenz 63 system acts as a forcing for the Lorenz 84 system, which represents the dynamics of interest. The dynamics of the two systems has a time scale separation given by the factor $\tau$ and can be written as follows:

$$\frac{dX}{dt} = -Y^2 - Z^2 - aX + a(F_0 + hx), \tag{7}$$

$$\frac{dY}{dt} = XY - bXZ - Y + G, \tag{8}$$

$$\frac{dZ}{dt} = XZ + bXY - Z, \tag{9}$$

$$\frac{d\widetilde{x}}{dt} = \tau s(\widetilde{y} - \widetilde{x}), \tag{10}$$

$$\frac{d\widetilde{y}}{dt} = \tau(\rho\widetilde{x} - \widetilde{y} - \widetilde{x}\widetilde{z}), \tag{11}$$

$$\frac{d\widetilde{z}}{dt} = \tau(-\beta\widetilde{z} + \widetilde{x}\widetilde{y}). \tag{12}$$

It is important to underline that the coupling between the Lorenz 84 and the Lorenz 63 is uni-directional: the latter model affects the former and, acts as an external forcing, with no feedback acting the other way around.

In what follows, we choose fairly classical values for the parameters: $a = 0.25$, $b = 4$, $s = 10$, $\rho = 28$, $\beta = 8/3$; the two forcings are set as $F_0 = 8$ (corresponding to the so-called winter conditions) and $G = 1$. The parameter $h$ is a modulation coefficient that defines the coupling strength and we choose $h = 0.25$ in order to provide a stochastic forcing between two and four orders of magnitude smaller (on average) than the tendencies of the $X$ variable (see below).

The parameter $\tau$ defines the ratio between the internal time scale of the two systems: in case of $\tau > 1$, the Lorenz 63 provides a forcing that is typically on time scales shorter than those of the system of interest; while if $\tau < 1$ the forcings can be interpreted as a modulating factor of the dynamics of the Lorenz 84 model. In the first case, in particular, we can interpret the Lorenz 63 as being the cause of the forcing exerted by convective motions in the synoptic scale dynamics described by the Lorenz 84 model. The numerical integration scheme used is a Runge-Kutta 4 with a time step of $0.005$ (Bódai et al., 2011).

Henceforth, we will refer to the standard Lorenz 84 as *uncoupled model*, whilst the Lorenz 84 subject to the coupling with the Lorenz 63 will be called *coupled model*.

## 3   Wouters Lucarini's parametrization

Wouters and Lucarini (2012, 2013, 2016) presented a top-down method suitable for constructing parametrizations for chaotic dynamical systems in the form:

$$\frac{d\mathbf{K}}{dt} = \mathbf{F_K}(\mathbf{K}) + \epsilon\mathbf{\Psi_K}(\mathbf{K},\mathbf{J}), \tag{13}$$

$$\frac{d\mathbf{J}}{dt} = \mathbf{F_J}(\mathbf{J}) + \epsilon\mathbf{\Psi_J}(\mathbf{K},\mathbf{J}), \tag{14}$$

where the $\mathbf{K} = (X, Y, Z)$ is the vector of the variables we are interested in and the $\mathbf{J} = (\tilde{x}, \tilde{y}, \tilde{z})$ is the vector of the variables we want to parametrize. The coefficient $\epsilon$ controls the strength of the couplings, i.e. $\mathbf{\Psi_K}(\mathbf{K},\mathbf{J})$ and $\mathbf{\Psi_J}(\mathbf{K},\mathbf{J})$.

The parametrization is obtained assuming the chaotic reference and applying Ruelle response theory (Ruelle, 1998, 2009); the effect of the coupling in Eq.(13) is approximated, up to the second order in $\epsilon$, by three terms: the first order consists in a deterministic term, while the second order includes a stochastic forcing and a non-markovian term. The general form of the parametrization (e.g. Vissio and Lucarini (2018)) is:

$$\frac{d\mathbf{K}}{dt} = \mathbf{F_K}(\mathbf{K}) + \epsilon\mathbf{D}(\mathbf{K}) + \epsilon\mathbf{S}\{\mathbf{K}\} + \epsilon^2\mathbf{M}\{\mathbf{K}\}, \tag{15}$$

where $\mathbf{D}$, $\mathbf{S}$ and $\mathbf{M}$ indicate, respectively, the deterministic, stochastic and memory terms and are defined below in Eqs.(18)-(22). Note that the projection on the variables of interest of invariant measure of the full system given in Eqs.(13)-(14) and the invariant measure of the system give in Eq.(15) are the same up to second order in the coupling parameter $\epsilon$, as discussed in Wouters and Lucarini (2013); Vissio and Lucarini (2018). Since the couplings are seen as a perturbation applied to an otherwise uncoupled system, the three terms in Eq.(15) can be calculated considering the statistical properties of the unperturbed

equations

$$\frac{d\mathbf{K}}{dt} = \mathbf{F_K}(\mathbf{K}),\tag{16}$$

$$\frac{d\mathbf{J}}{dt} = \mathbf{F_J}(\mathbf{J}).\tag{17}$$

The numerical integration of Eqs.(16)-(17) may allow to use less computational resources with respect to Eqs.(13)-(14), par-
ticularly in the case of multiscale systems.

As discussed in (Vissio and Lucarini, 2018), WL parametrization has the remarkable feature of having a good degree of adaptivity in terms of changes to the time scale separation between the $\mathbf{K}$ and $\mathbf{J}$ variables, to be performed by rescaling, e.g. $t \to \tau$ in Eq.(17). In this scale, the term $D(K)$ in Eq.(15) is unchanged, while the time scale of the autocorrelation of the noise term $S(K)$ and of the memory term $M(K)$ are reduced by a factor $\tau/t$. In the specific case of the Lorenz 96 system studied in Vissio and Lucarini (2018), the adaptivity is more general than the one related to changes in the time scale separation only, and points to the possibility of developing general adaptive parametrization schemes beyond such specific model. It is not yet clear whether this might lead to constructing spatial scale-adaptive parametrizations.

## 3.1 Constructing the parametrization

The coupling strength $\epsilon$, shown in Eqs.(13)-(14) and in Eq.(15), assumes the value $\epsilon = ah$, while the couplings are, with respect to the vector $(X, Y, Z)$ in Lorenz 84 phase space, $\mathbf{\Psi_K}(\mathbf{K}, \mathbf{J}) = \mathbf{\Psi_K}(\mathbf{J}) = (\widetilde{x}, 0, 0)$ and $\mathbf{\Psi_J}(\mathbf{K}, \mathbf{J}) = \mathbf{\Psi_J}(\mathbf{K}) = (0, 0, 0)$. Note that this is a case of independent coupling - i.e. a coupling that depends only on the variable of the other equation -, for which the application of the methodology is simpler than the dependent coupling case (Wouters and Lucarini, 2012).

The deterministic term $\mathbf{D}$ in Eq.(15) is a measure of the average impact of the coupling on the $\mathbf{K}$ dynamics and can be written as:

$$\mathbf{D}(\mathbf{K}) = \rho_{0,\mathbf{J}}(\mathbf{\Psi_K}(\mathbf{J})) = \lim_{T \to \infty} \frac{1}{T} \int_0^T \mathbf{\Psi_K}(\mathbf{J}) d\tau = \rho_{0,\mathbf{J}}((\tilde{x}), 0, 0) = \lim_{T \to \infty} \frac{1}{T} \int_0^T (\widetilde{x}(\tau), 0, 0) d\tau = (D, 0, 0),\tag{18}$$

where $\rho_{0,\mathbf{x}}(A)$ ($\mathbf{x} = \mathbf{K}, \mathbf{J}$) is the expectation value of $A$ computed according to the invariant measure given by the uncoupled dynamics of the $\widetilde{x}$ variables and we have used ergodic averaging. We have used the expression of the coupling given in Eq.(7) and we have computed the ensemble average as time average on the ergodic measure of $\widetilde{x}$. Since the measure of Lorenz 63 is symmetric for $\widetilde{x} \to -\widetilde{x}$, one could think of choosing $\mathbf{D}(\mathbf{K}) = (0, 0, 0)$. Nevertheless, this is the limit for a run of infinite time length, while in a numerical simulation we must choose a finite number of steps - in our case $146000$, ten years in Lorenz models. Therefore, it seems appropriate to calculate $\mathbf{D}$ using the time series given by the uncoupled Lorenz 63 and Eq.(18), as we do for the second order of the parametrization, see below.

Since the coupling shown in Eq.(7) depends only on one of the variables (in this case the $\widetilde{x}$) of the system we want to parametrize, the stochastic term can be written as

$$\mathbf{S}\{\mathbf{K}\} = (\omega(t), 0, 0),\tag{19}$$

where the properties of $\omega(t)$, a stochastic noise, are defined by its correlation $R(t)$ and its time average $\langle \omega(t) \rangle$:

$$\mathbf{R(t)} = \langle (\omega(0),0,0),(\omega(t),0,0)\rangle = \rho_{0,\mathbf{J}}((\mathbf{\Psi_K(J)} - \mathbf{D(K)})(\mathbf{\Psi_K}(\mathbf{f}^t(\mathbf{J})) - \mathbf{D(K)})),$$
$$= \rho_{0,\mathbf{J}}(((\widetilde{x}(0),0,0) - (D,0,0))((\widetilde{x}(t),0,0) - (D,0,0))),$$
$$\langle \omega(t) \rangle = 0. \tag{20}$$

As discussed in Wouters and Lucarini (2012, 2013); Vissio and Lucarini (2018), for more complex couplings the stochastic term assumes the form of a multiplicative noise. We have used the software package ARFIT (Neumaier and Schneider, 2001; Schneider and Neumaier, 2001) to construct time series of noise with the desired properties defined by Eq.(20).

The last term in Eq.(15) is the non-markovian contribution to the parametrization and can be written as follows:

$$\mathbf{M\{K\}} = \int_0^\infty \mathbf{h}(t_2, \mathbf{K}(t-t_2))dt_2, \tag{21}$$

where

$$\mathbf{h}(t_2, \mathbf{K}) = \mathbf{\Psi_J(K)}\rho_{0,\mathbf{J}}(\partial_{\mathbf{J}}\mathbf{\Psi_K}(\mathbf{f}^{t_2}(\mathbf{J}))) = (0,0,0) \cdot \rho_{0,\mathbf{J}}(\partial_{\mathbf{J}}(\widetilde{x}(f^{t_2}(\widetilde{x},0,0)),0,0)). \tag{22}$$

As discussed in Section 2.3, the evolution of the variables of the Lorenz 63 model - see Eqs.(10)-(12) - are independent of the state of the variables corresponding to the Lorenz 84 model. As a result, the first factor on the r.h.s. of Eq.(22) vanishes, so that the parametrization we derive is fully markovian.

After the implementation of Wouters-Lucarini's procedure, Eq.(7) will be parametrized as

$$\frac{dX}{dt} = -Y^2 - Z^2 - aX + a[F_0 + h(D+S)]; \tag{23}$$

Eq.(23), together with Eqs.(8)-(9), will be henceforth indicated as the system constructed with *second order parametrization*, whilst the same equations without the stochastic term (therefore comprehending the first order, deterministic term only), namely

$$\frac{dX}{dt} = -Y^2 - Z^2 - aX + a[F_0 + hD], \tag{24}$$

will be called *first order parametrization*.

## 4  Wasserstein Distance

We wish to assess how well a parametrization allows to reproduce the statistical properties of the full coupled system. At this regard, it seems relevant to quantify to what extent the projected invariant measure of the full coupled model on the variables of interest differs from the invariant measures of the surrogate models containing the parametrization. In order to evaluate how much such measures differ, we resort to considering their Wasserstein distance (Villani, 2009). Such a distance quantifies

the minimum "effort" in morphing one measure into the other, and was originally introduced by Monge (1781), somewhat unsurprisingly, to study problems of military relevance, and later improved by Kantorovich (1942).

Starting from two distinct spatial distribution of points, described by the measures $\mu$ and $\nu$, we can define the optimal transport cost (Villani, 2009) as the minimum cost to move the set of points corresponding to $\mu$ into the set of points corresponding to $\nu$:

$$C(\mu, \nu) = \inf_{\pi \in \Pi(\mu, \nu)} \int c(x, y) d\pi(x, y), \tag{25}$$

where $c(x, y)$ is the cost for transporting one unit of mass from $x$ to $y$ and $\Pi(\mu, \nu)$ is the set of all joint probability measures whose marginals are $\mu$ and $\nu$. The function $C(\mu, \nu)$ in Eq.(25) is called Kantorovich-Rubinstein distance. In the rest of the paper, we will consider the Wasserstein distance of order 2:

$$W_2(\mu, \nu) = \left\{ \inf_{\pi \in \Pi(\mu, \nu)} \int [d(x, y)]^2 d\pi(x, y) \right\}^{\frac{1}{2}}. \tag{26}$$

We can define the Wasserstein distance also in the case of two discrete distributions

$$\mu = \sum_{i=1}^{n} \mu_i \delta_{x_i}, \tag{27}$$

$$\nu = \sum_{i=1}^{n} \nu_i \delta_{y_i}, \tag{28}$$

where $x_i$ and $y_i$ represent the location of the different points, which mass is given, respectively, by $\mu_i$ and $\nu_i$. Recalling the definition of Euclidean distance

$$d(\mu, \nu) = \left[ \sum_{i=1}^{n} (x_i - y_i)^2 \right]^{\frac{1}{2}}, \tag{29}$$

we can construct the order 2 Wasserstein distance for discrete distributions as follows:

$$W_2(\mu, \nu) = \left\{ \inf_{\gamma_{ij}} \sum_{i,j} \gamma_{ij} [d(x_i, y_j)]^2 \right\}^{\frac{1}{2}}. \tag{30}$$

where $\gamma_{ij}$ is the fraction of mass transported from $x_i$ to $x_j$.

This latter definition of Wasserstein distance has already been proven effective (Robin et al., 2017) for providing a quantitative measurement of the difference between the snapshot attractors of the Lorenz 84 system in the instance of summer and winter forcings.

Hereby we propose to further assess the reliability of WL stochastic parametrization by studying the Wasserstein distance between the projected invariant measure of the original system on the first three variables $(X, Y, Z)$ and the invariant measures obtained using the surrogate dynamics corresponding to the first and second order parametrization. Nevertheless, since the numerical computations for optimal transport through linear programming theory are not cheap, a new approach is required. In order to accomplish it, we perform a standard Ulam discretization (Ulam, 1964; Tantet et al., 2018) of the measure supported

on the attractor. By coarse-graining on a set of cubes with constant sides across the phase space. We will discuss below the impact of changing the sides of such cubes.

The coordinates of the cubes will then be equal to the location $x_i$, while the corresponding densities of the points are used to define $\gamma_{ij}$; finally, we exclude from the subsequent calculation all the grid boxes containing no points at all.

Our calculations are performed using a modified version of the software for Matlab written by Gabriel Peyré and made available at http://www.numerical-tours.com/matlab/optimaltransp_1_linprog/, conveniently modified to include the subdivision of the phase space in cubes and the assignment of corresponding density to those cubes.

## 5 Parametrizing the Coupling with the Lorenz 63 Model

In this section we show the results corresponding to the case $\tau = 5$. Therefore, Lorenz 84 and Lorenz 63 are seen as, respectively, the slow and the fast dynamical systems.

### 5.1 Qualitative Analysis

We first provide a qualitative overview of the performance of the parametrization by investigating a few Poincaré sections, which provide a convenient and widely used method to visualize the dynamics of a system in a two-dimensional plot (Eckmann and Ruelle, 1985; Ott, 1993); typically, the plane chosen for the section of Lorenz 84 is $Z = 0$. Fig.1a) shows the Poincaré section at $Z = 0$ of the variables $X$, $Y$ of the coupled model given in Eqs.(7)-(12). Panels b) of the same figure shows the Poincaré section of the Lorenz 84 model obtained by removing the coupling with the Lorenz 63 model. Finally, Panels c) and d) show the Poincaré sections of the modified Lorenz 84 models obtained by adding the first and second order parametrization, respectively. Visual inspection suggests that the second order parametrization does a good job in reproducing the properties of the full coupled model.

Metaphorically, our parametrization aims at describing as accurately as possible the impact of "convection" on the "westerlies". It is insightful to look at how it affects the properties of the two variables - $Y$ and $Z$ - that are not directly impacted by it. This amounts to looking at the impact of the parametrization of "convection" on the "large scale planetary waves" and, consequently, on the "large scale heat transport". Therefore, we look into $X = constant$ Poincaré section, in order to highlight the properties of $Y$ and $Z$. The four panels in Fig.2 are structured as in Fig.1 and depict the Poncaré section of $X = 1$. Also in this case the second order parametrization provides a far better match to the coupled model, featuring a remarkable ability in the reproducing the main features of the pattern of points.

In order to provide further qualitative evidence of our results, in four panels of Fig.3 we show the trajectories in the phase space of the $X$, $Y$, and $Z$ variables for the four considered models. For the sake of clarity, the plots are created using just 5 years (365 time units). In the case of the coupled model the attractor spans over more extreme values of the variables and the second order parametrization successfully imitates this feature, while the simple deterministic correction, once again, is completely inadequate.

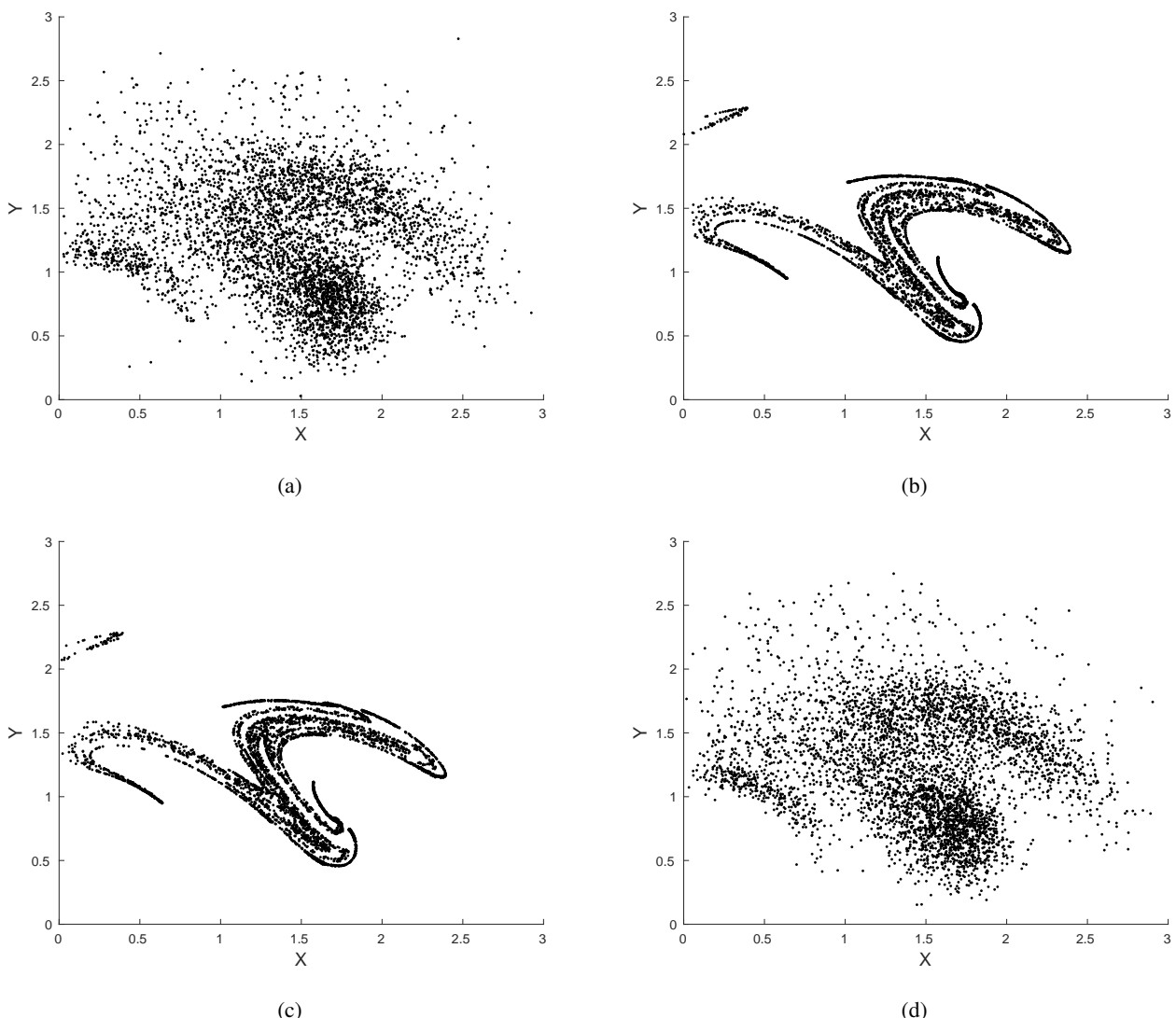

**Figure 1.** Poincaré section in $Z = 0$ of a) coupled model; b) uncoupled model; c) 1st order parametrization; d) 2nd order parametrization. Case $\tau = 5$, the Lorenz 63 model acts as a fast forcing on the Lorenz 84 model.

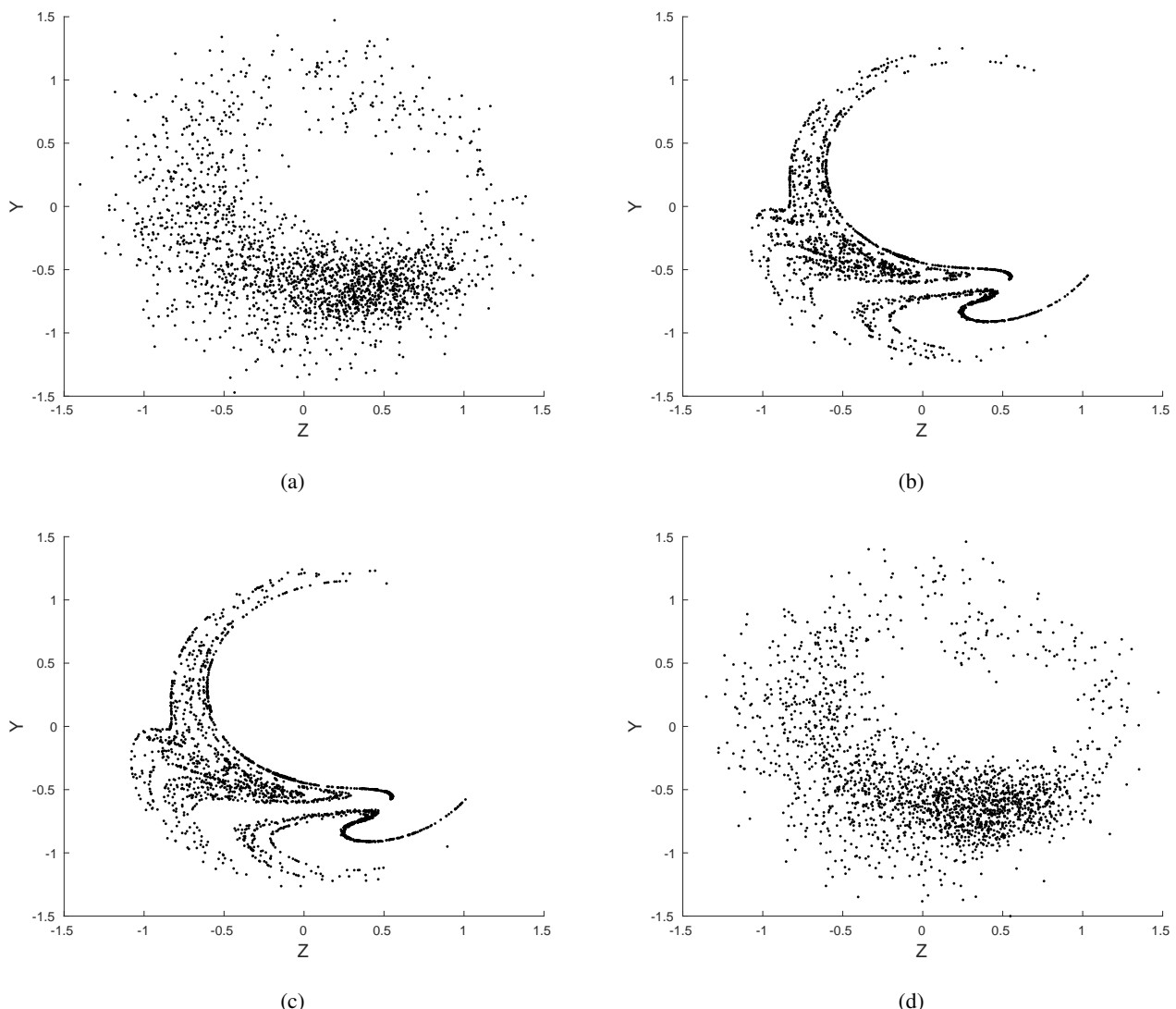

**Figure 2.** Poincaré section in $X = 1$ of a) coupled model; b) uncoupled model; c) 1st order parametrization; d) 2nd order parametrization. Case $\tau = 5$, the Lorenz 63 model acts as a fast forcing on the Lorenz 84 model.

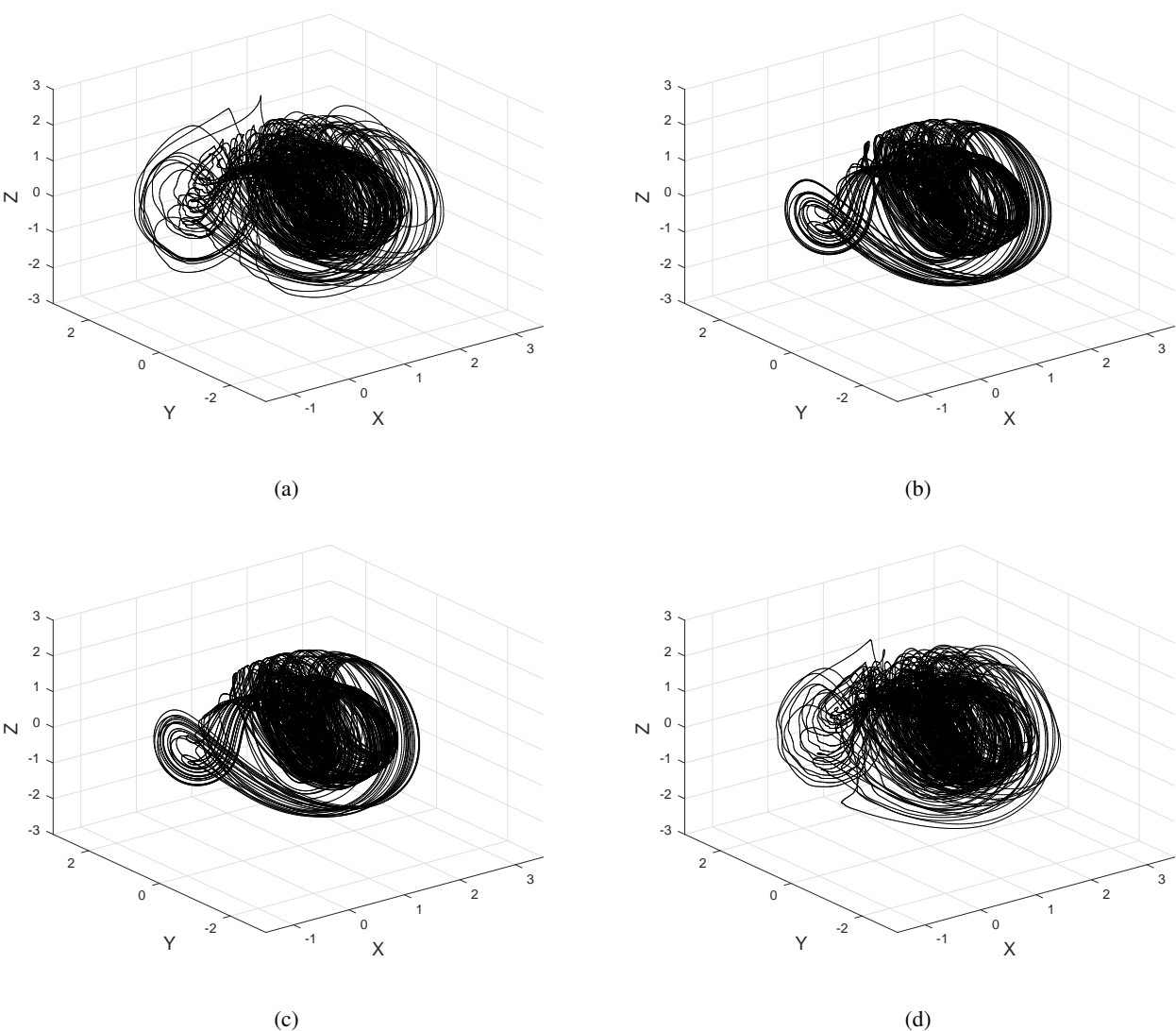

(a)

(b)

(c)

(d)

**Figure 3.** $3D$ view of the attractor of a) coupled model; b) uncoupled model; c) 1st order parametrization; d) 2nd order parametrization. Case $\tau = 5$, the Lorenz 63 model acts as a fast forcing on the Lorenz 84 model.

## 5.2 Evaluation of the Performance of the Parametrization

Further to the qualitative inspection, we provide here quantitative comparisons to support our study. All the remaining simulations in this section are run for 100 years (7300 time units) with a time step of 0.005; thus, each attractor is constructed with 1460000 points. We have tested that the results presented below are virtually unchanged when considering a smaller time step of 0.001.

We first look into the probability densities (PDFs) of the variables $X$, $Y$ and $Z$, which describe, loosely speaking, our climate. Fig.4 shows the PDF of the $X$ variable for the four considered models. As expected, the second order parametrization allows for reconstructing with great accuracy the statistics of the coupled model. The bimodality of the uncoupled Lorenz 84 model is reproduced by the model featuring the first order parametrization, while the second order model predicts accurately the unimodal distribution shown by the coupled model. The PDFs for $Y$ and $Z$ variables are shown in Figs.5-6, respectively. Also here, where the external forcing does not destroy the bimodality of the distributions found in the uncoupled case, WL parametrization leads to a very good approximation of the properties of the coupled model. In particular, the tails of the distributions are represented with a high level of precision, making possible to seemingly reproduce with good accuracy the extreme values of the variables. This is a matter worth investigating in a separate study. Note that, since the WL parametrization is constructed to have skill for all observables, it is not so surprising that it can perform well also far away from the bulk of the statistics, see discussion in Lucarini et al. (2014b).

Aside from the analysis of the PDF, a further statistical investigation can be provided by looking into the numerical results provided by first moments of the variables and their uncertainty, which is computed as the standard deviation derived from the analysis of an ensemble of runs. We have performed just ten runs, but our results are very robust. The results for the statistics of the first two moments are reported in Table 1: all the quantities inspected clearly show that the second order parametrization allows for reproducing very accurately the moments statistics of the coupled model.

If the considered PDFs depart strongly from uni-modality, the analysis of the first moments can be of little utility, and it becomes hard to have a thorough understanding of the statistics by adopting this point of view. As discussed above, we wish to supplement this simple analysis with a more robust evaluation of the performance of the parametrizations by taking into account the Wasserstein distance. A first issue to deal with in order compute the Wasserstein distance consists in carefully choosing the number of cubes used for the Ulam projection. Fig.7a shows the Wasserstein distance between the invariant measure of the coupled model projected on the $XYZ$ space and the invariant measure of the uncoupled Lorenz and of the models obtained using the first and second order parametrization. We find that for all choices of the coarse-graining the measure of the model with the second order parametrization is, by far, the closest to the projected measure of the coupled model. Instead, the deterministic parametrization provides a negligible improvement with respect to the trivial case of considering the uncoupled model, as expected given the discussion following Eq.(18). What shown here gives a quantitative evaluation of the improved performance resulting from adding a stochastic parametrization. The second piece of information is that the estimated Wasserstein distance has only a weak dependence on the degree of the coarse-graining and seems to approach its

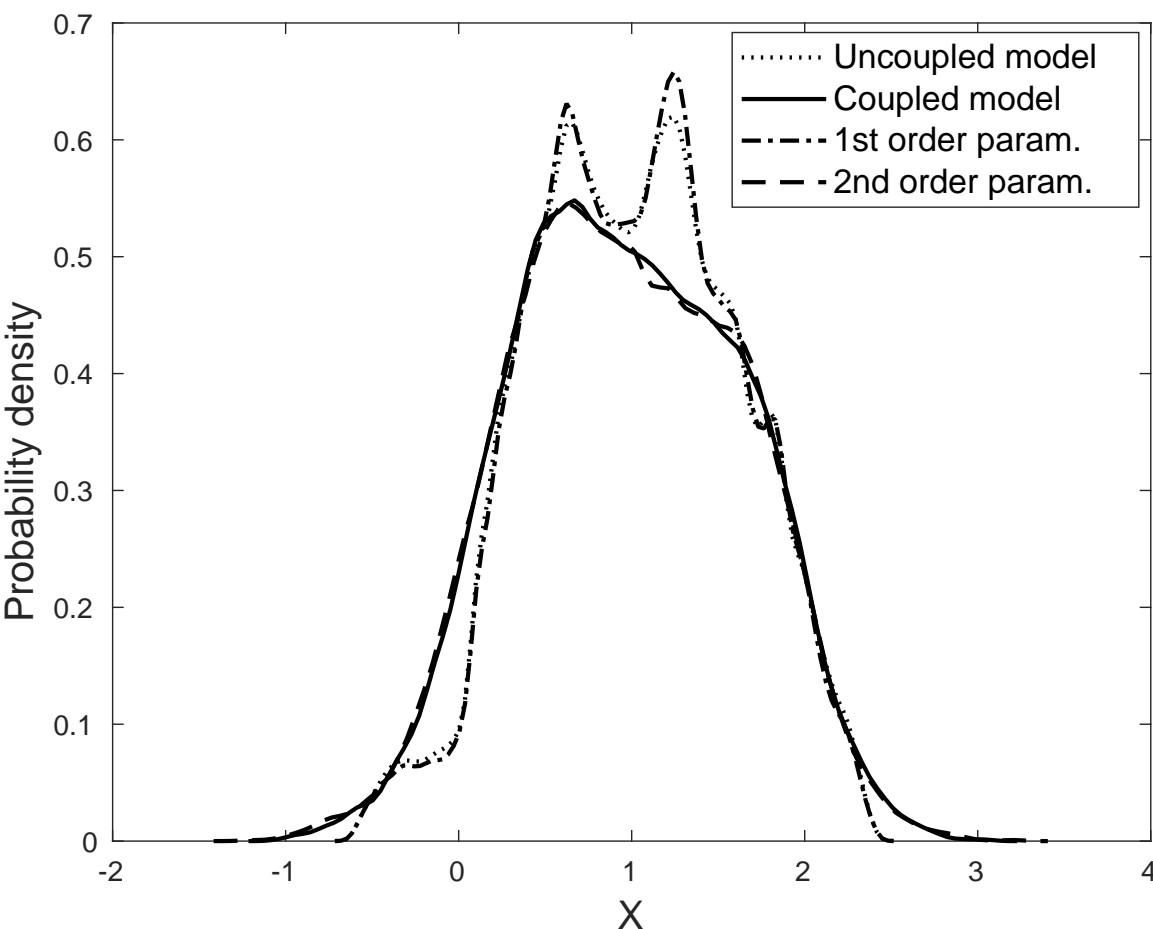

**Figure 4.** Probability density of the $X$ variable. Case $\tau = 5$, the Lorenz 63 model acts as a fast forcing on the Lorenz 84 model.

asymptotic value for the finest (yet still pretty coarse) Ulam partitions considered here. This is encouraging as the findings one can obtain at low resolution seem to be already very meaningful and useful.

A well-known problem of Ulam's method is the fact that it can hardly be adapted to high dimensional spaces - this is the so-called curse of dimensionality. Additionally, evaluating the Wasserstein distance in high dimensions becomes itself
5  computationally extremely challenging. In order to partially address these problems we repeat the analysis shown in Fig.7a) for the measures projected on the $XY$, $XZ$ and $YZ$ planes, thus constructing the so-called sliced Wasserstein distances. Results are reported in panels b), c), and d) of Fig.7, respectively. We find that, unsurprisingly, the distance of the projected measure is strictly lower than the distance in the full phase space, *ceteris paribus*. What is more interesting is that all the observations we

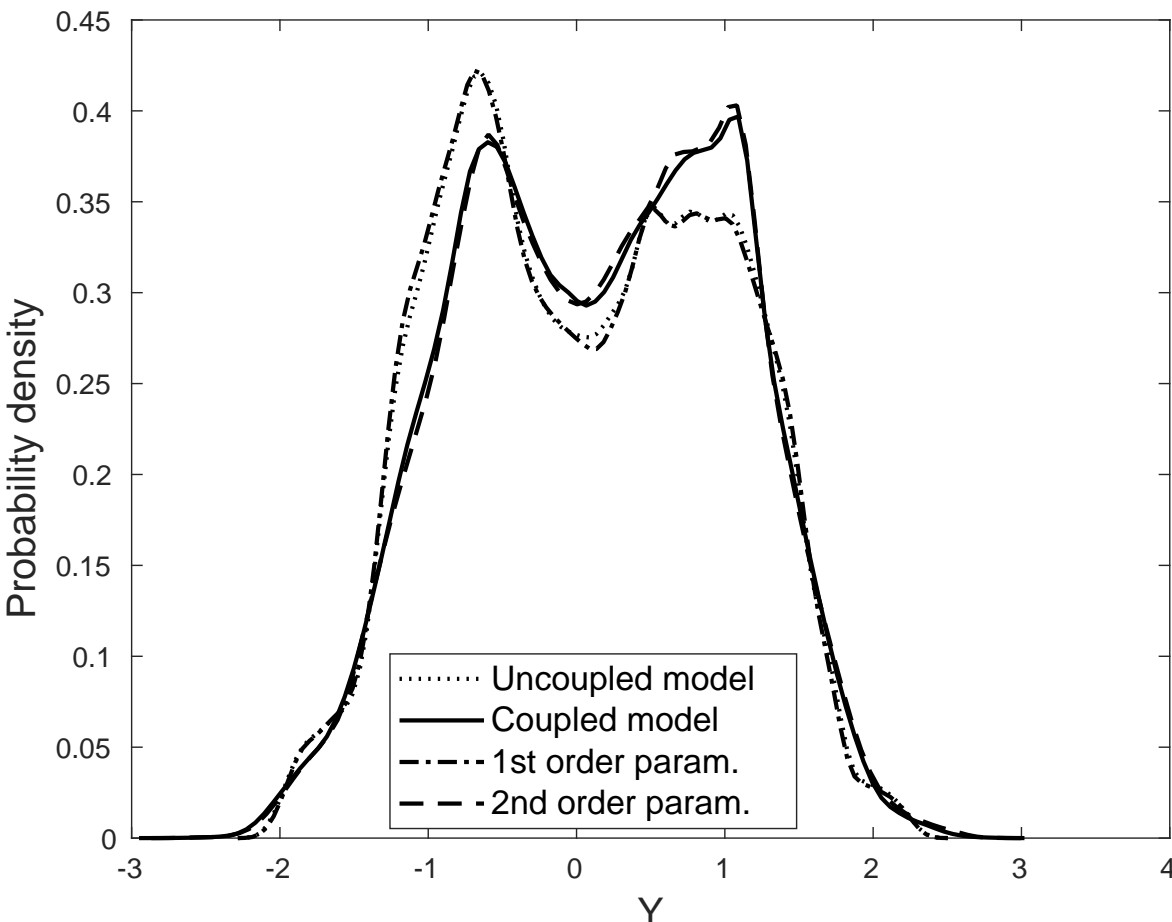

**Figure 5.** Probability density of the $Y$ variable. Case $\tau = 5$, the Lorenz 63 model acts as a fast forcing on the Lorenz 84 model.

made for Fig.7a) apply for the other panels. Therefore, it seems reasonable to argue that studying the Wasserstein distance for projected spaces might provide useful information also on the full system.

In order to extend the scope of our study we have repeated the analysis described above for the case $\tau = \frac{1}{6}$. Such a choice implies that the model responsible for the forcing has an internal time scale which is larger than the one of the model of interest. We remark that the WL parametrization, as discussed in (Vissio and Lucarini, 2018), is not based on any assumption of time scale separation between the variables of interest and the variables we want to parametrize. We report below only the main results for the sake of conciseness.

Figures 8a)-d) show the Poincaré sections in $Z = 0$ for all the considered models. In the case of the coupled system, most of the fine structure one finds in the uncoupled model is lost, and we basically have a cloud of points with weaker features than

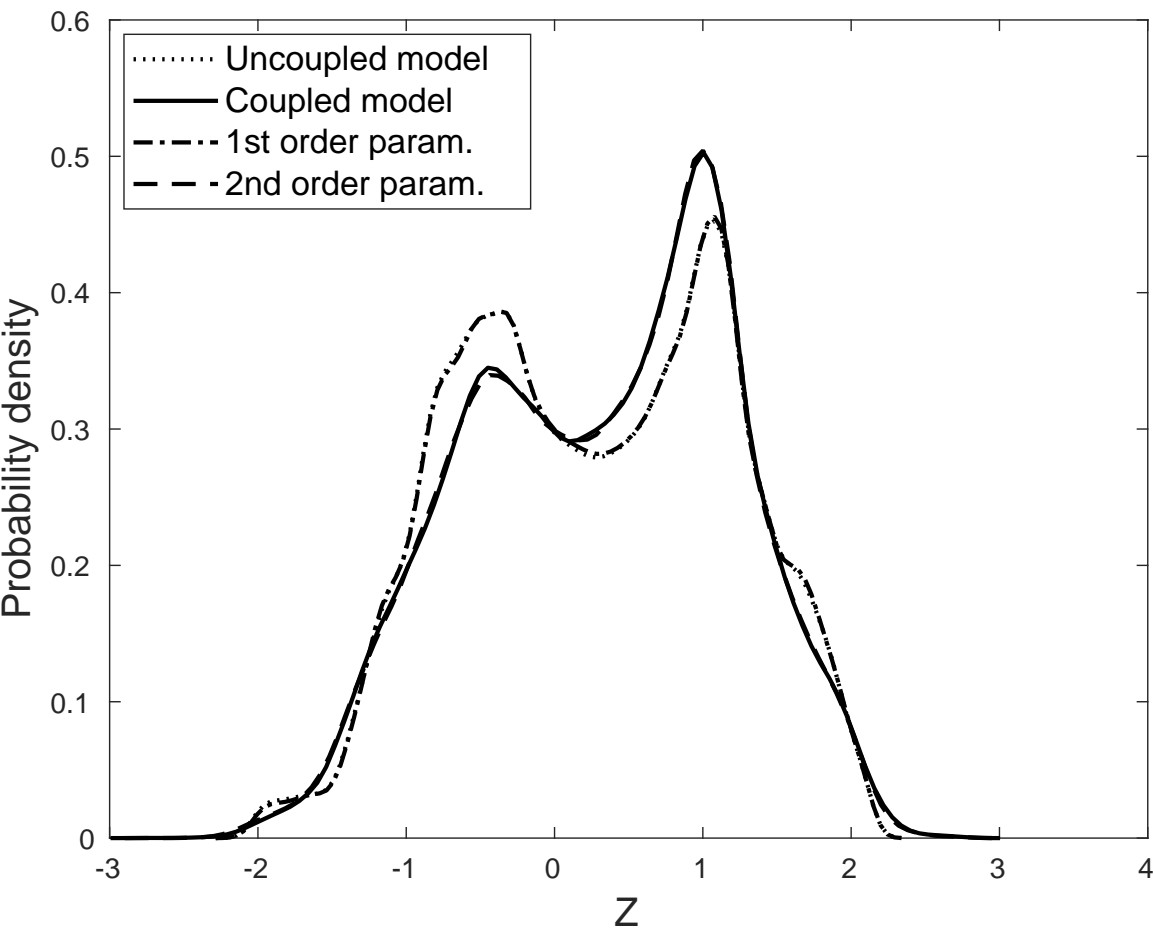

**Figure 6.** Probability density of the $Z$ variable. Case $\tau = 5$, the Lorenz 63 model acts as a fast forcing on the Lorenz 84 model.

what shown in Figure 1 for $\tau = 5$. Nonetheless, also in this case the model with the second order parametrization reproduces (visually) quite well what shown in Panel a), and, in particular, shows matching regions where the density of the points is higher.

The analysis performed considering the Wasserstein distance between the measures is shown in Fig.9. Without going into

5 details, one finds that the same considerations we made for $\tau = 5$ are still valid for $\tau = \frac{1}{6}$ regarding the performance of the parametrization schemes and the role of coarse graining. Additionally, we observe that, for each choice of coarse-graining, the distance between the measure of the parametrized models and the actual projected measure of the coupled model is larger for $\tau = \frac{1}{6}$, thus indicating the parametrization procedure performs worse in this case. This fits with the intuition one can have by checking out how well Panels b)-d) reproduce Panel a) in Fig. 8 versus the case of Fig. 1.

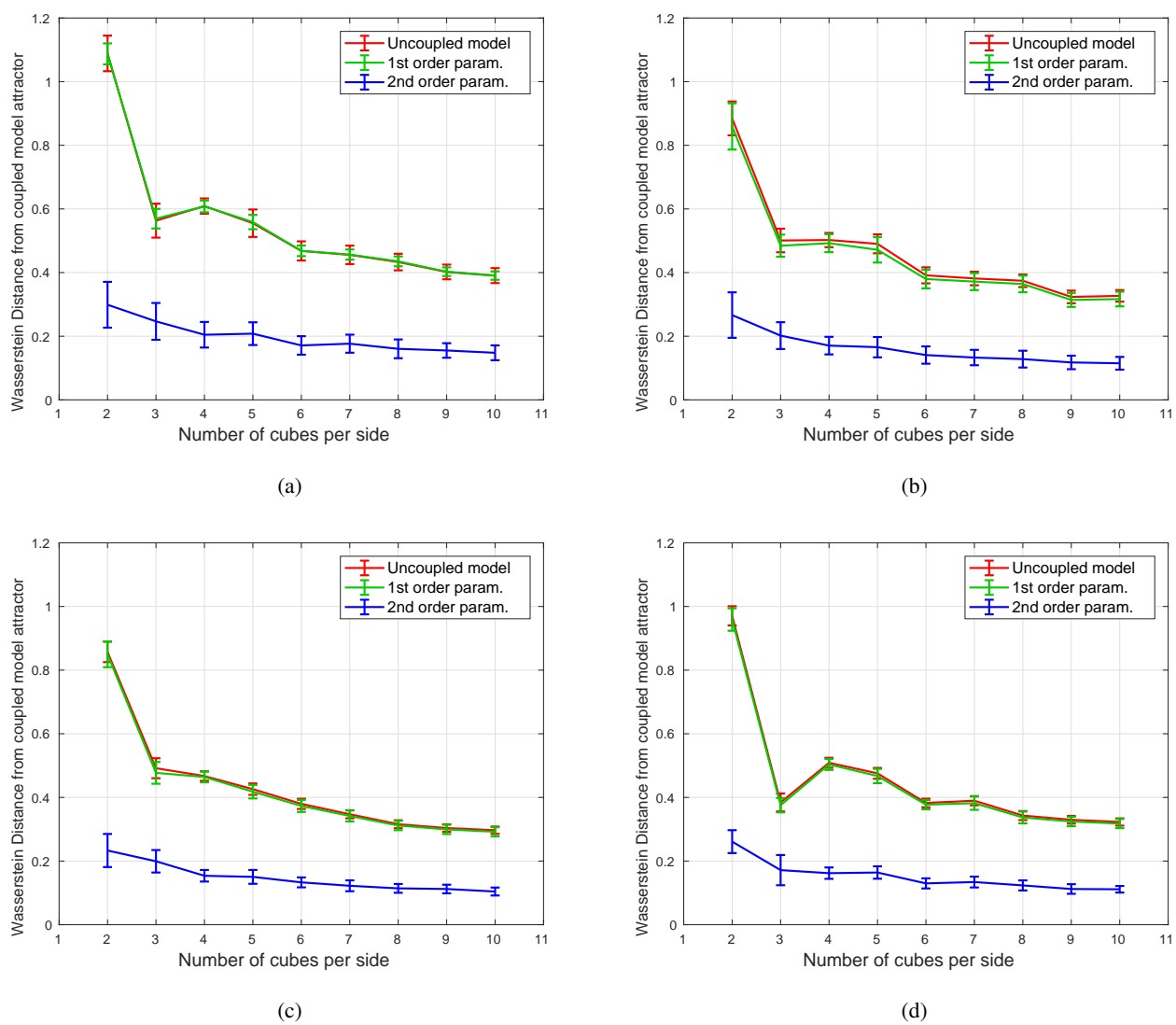

**Figure 7.** Wasserstein distances from the coupled model with respect to number of cubes per side: a) $3D$ case; b) Projection on $XY$ plane; c) Projection on $XZ$ plane; d) Projection on $YZ$ plane. Case $\tau = 5$, the Lorenz 63 model acts as a fast forcing on the Lorenz 84 model.

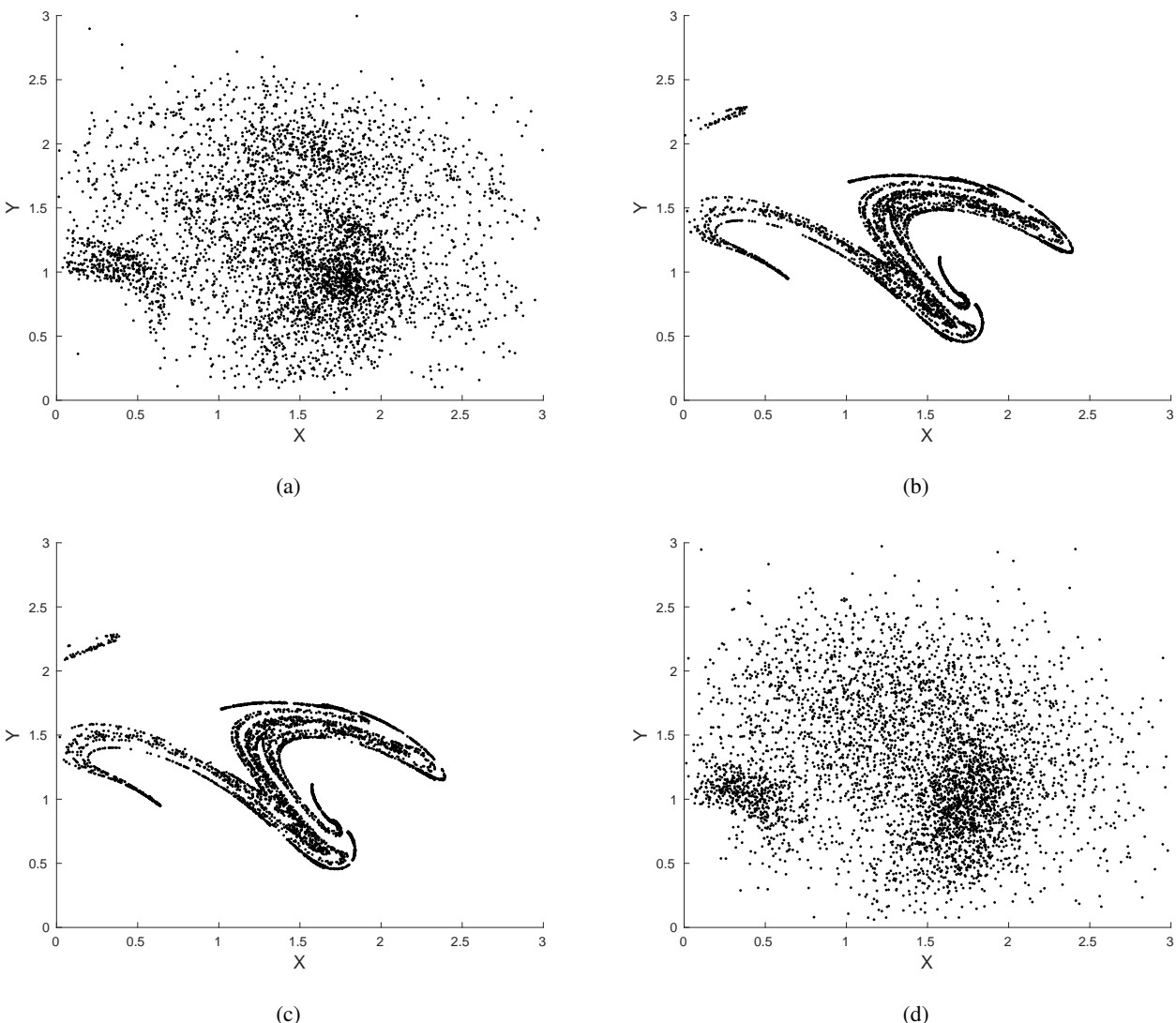

**Figure 8.** Poincaré section in $Z = 0$ of a) coupled model; b) uncoupled model; c) 1st order parametrization; d) 2nd order parametrization. Case $\tau = \frac{1}{6}$, the Lorenz 63 model acts as a slow forcing on the Lorenz 84 model.

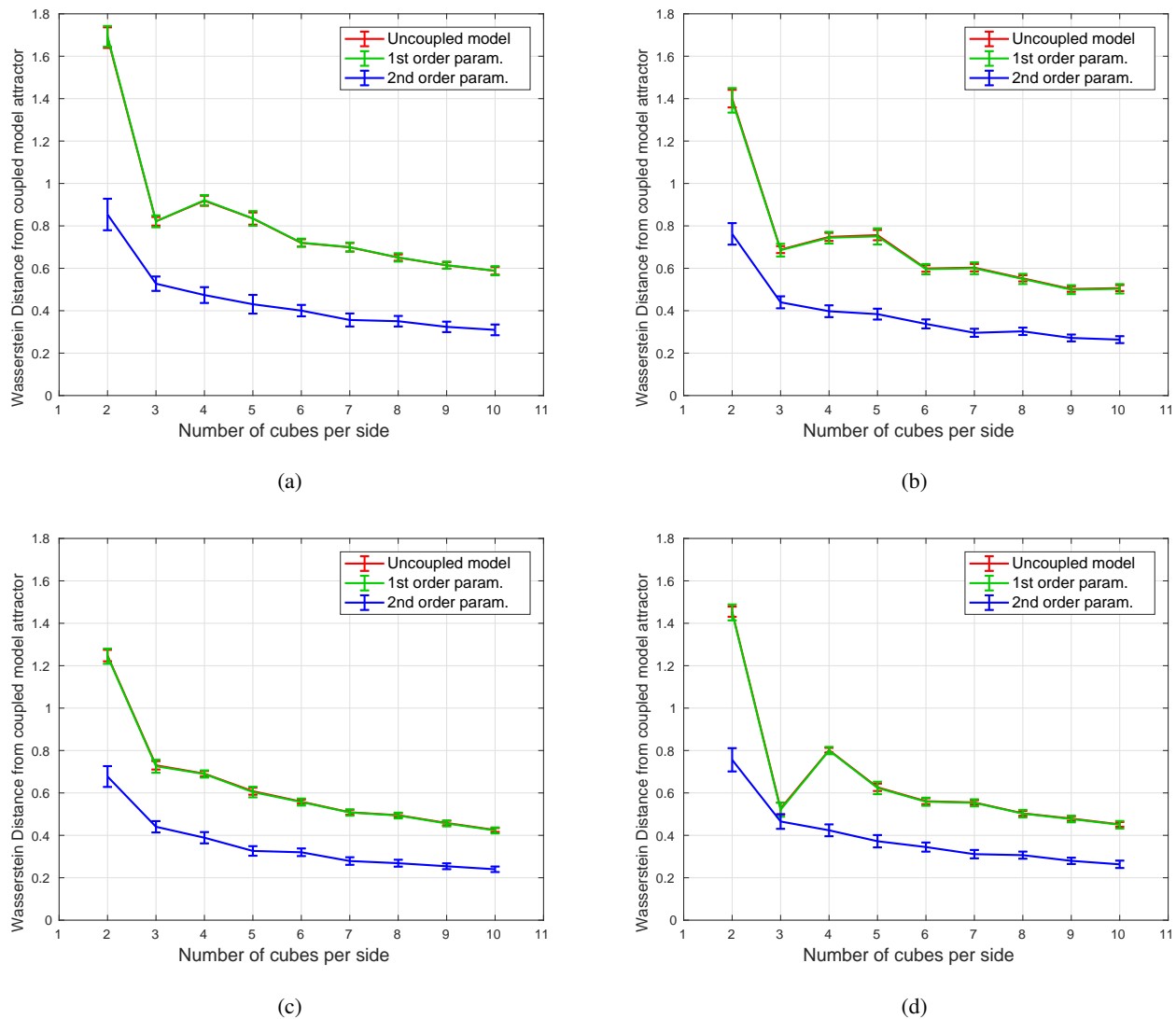

**Figure 9.** Wasserstein distances from the coupled model with respect to number of cubes per side: a) $3D$ case; b) Projection on $XY$ plane; c) Projection on $XZ$ plane; d) Projection on $YZ$ plane. Case $\tau = \frac{1}{6}$, the Lorenz 63 model acts as a slow forcing on the Lorenz 84 model.

**Table 1.** Expectation values for the ensemble average of the first two moments of the variables $X$, $Y$, and $Z$. The uncertainty is indicated as standard deviation (std) over the ensemble of realizations. with the corresponding standard deviations $\sigma$. All the values are multiplied by $10^2$. Case $\tau = 5$, Lorenz 63 as fast scale model.

| Observables | Uncoupled model ($\times 10^2$) | 1st order parametrization ($\times 10^2$) | 2nd order parametrization ($\times 10^2$) | Coupled model ($\times 10^2$) |
|---|---|---|---|---|
| $\overline{X} \pm \sigma_{\overline{X}}$ | $101.5 \pm 0.4$ | $101.3 \pm 0.5$ | $97.2 \pm 0.3$ | $97.1 \pm 0.3$ |
| $\overline{Y} \pm \sigma_{\overline{Y}}$ | $6.1 \pm 0.8$ | $6.5 \pm 1.2$ | $13.7 \pm 0.7$ | $13.9 \pm 0.4$ |
| $\overline{Z} \pm \sigma_{\overline{Z}}$ | $27.0 \pm 0.2$ | $26.9 \pm 0.3$ | $31.0 \pm 0.2$ | $31.3 \pm 0.5$ |
| $var(X) \pm \sigma_{var(X)}$ | $34.9 \pm 0.8$ | $35.2 \pm 1.0$ | $43.6 \pm 0.7$ | $43.5 \pm 0.3$ |
| $var(Y) \pm \sigma_{var(Y)}$ | $84.4 \pm 0.1$ | $84.4 \pm 0.1$ | $82.8 \pm 0.4$ | $82.6 \pm 0.3$ |
| $var(Z) \pm \sigma_{var(Z)}$ | $82.6 \pm 0.1$ | $82.6 \pm 0.2$ | $81.5 \pm 0.3$ | $81.4 \pm 0.3$ |
| $cov(XY) \pm \sigma_{cov(XY)}$ | $-5.4 \pm 0.8$ | $-5.7 \pm 1.1$ | $-11.1 \pm 0.6$ | $-11.2 \pm 0.3$ |
| $cov(XZ) \pm \sigma_{cov(XZ)}$ | $-3.7 \pm 0.1$ | $-3.4 \pm 0.2$ | $-8.0 \pm 0.2$ | $-8.3 \pm 0.4$ |
| $cov(YZ) \pm \sigma_{cov(YZ)}$ | $-7.7 \pm 0.2$ | $-7.7 \pm 0.4$ | $-1.6 \pm 0.4$ | $-1.3 \pm 0.2$ |

## 6 Conclusions

Developing parametrizations able to surrogate efficiently and accurately the dynamics of unresolved degrees of freedom is a central task in many areas of science, and especially in geosciences. There is no obvious protocol in testing parametrizations for complex systems, because one is bound to look only at specific observables of interests. This procedure is not error-free, because optimizing a parametrization against one or more observables might lead to unfortunate effects on other aspects of the system and worsen, in some other aspects, its performance.

In this paper we have addressed the problem of constructing a parametrization for a simple yet meaningful two-scale system, and then testing its performance in a possibly comprehensive way. We have considered a simple six-dimensional system constructed by coupling a Lorenz 84 system and a Lorenz 63 system, with the latter acting as forcing to the former, and the former being the subsystem of interest. We have included a parameter controlling the time scale separation of the two system and a parameter controlling the intensity of the coupling. We have built a first order and a second order parametrization able to surrogate the effects of the coupling using the scale-adaptive WL method. The second order scheme includes a stochastic term, which has proved to be essential for radically improving the quality of the parametrization with respect to the purely deterninic case (first order parametrization), as already visually shown by looking at suitable Poincaré sections.

We show here that, in agreement of what shown in previous papers, the WL-approach provides an accurate and flexible framework for constructing parametrizations. Nonetheless, the main novelty of this paper lies in our use of the Wasserstein distance as a comprehensive tool for measuring how different the invariant measures ("the climates") of the uncoupled Lorenz 84 model, and of its two version with deterministic and stochastic parametrizations are from the projection of the measure of the coupled model on the variables of the Lorenz 84 model. We discover that the Wasserstein distance provides a robust tool for assessing the quality of the parametrization, and, quite encouragingly, meaningful results can be obtained when considering very coarse grained representation of the phase space. A well-known issues of using a methodology like the Wasserstein

distance is the so-called curse of dimensionality: the procedure itself becomes unfeasible when the system has a number of degree of freedom above few units. We have addressed (partially) this issue by looking at the Wasserstein distance of the projected measures on the three two-dimensional spaces spanned by two of the three variables of the Lorenz 84 model. We find that the properties of the Wasserstein distance in the reduced spaces follow closely those found in the full space. We maintain

that diagnostics based on the Wasserstein distance in suitably defined reduced phase spaces should become standard in the analysis of the performance of parametrizations and in intercomparing models of any level of complexity.

**Acknowledgement**

The authors wish to thank the reviewers and the editor for providing constructive criticism to the paper, which has stimulated an improvement of the quality of the paper. The authors wish to thank G. Peyré for making the Matlab software related

to Wasserstein Distance publicly available. GV was supported by the Hans Ertel Center for Weather Research (HErZ), a collaborative project involving universities across Germany, the Deutscher Wetterdienst and funded by the BMVI (Federal Ministry of Transport and Digital Infrastructure, Germany, Grant Agreement number U4603BMV1501). VL acknowledges the financial support provided by the DFG SFB/Transregio Project TRR181 (Grant Agreement number U4603SFB160110) and by the Horizon2020 projects BlueAction (Grant Agreement number 727852) and CRESCENDO (Grant Agreement number

641816). VL wishes to thank M. Ghil for having suggested the relevance of the Wasserstein distance, and Robin et al. (2017) for having written a stimulating paper at this regard. VL recalls several fond memories of very informal yet enlightening discussions with A. Trevisan on nonlinear dynamics and data assimilation.

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
