# Peer review of "Evaluating a Stochastic Parametrization of the Coupling in a Fast-Slow System using the Wasserstein Distance"

_Nonlinear Processes in Geophysics, 2018_

## Referee Comment (RC1) · Anonymous Referee #1 · 23 Mar 2018

The authors are concerned with a system where an L84 system is weakly coupled with a L63 system, applying the recently introduced Wouters-Lucarini (WL) linear response approach to devise a reduced model for the L84-variables.

The authors show numerically that the reduced system is able to reproduce the mean and covariance of the resolved L84 variables of the full system. Furthermore they use the Wasserstein distance to assess the relative merit of a first order and a second order WL parametrization. They show that the WL parametrization provides a reliable parametrization in the case when the driving L63 is a fast driver and when it is slow driver. In particular the latter case is not amenable to classical homogenization theory.

I recommend publication subject to some minor issues:

1.) page 2: The Mori-Zwanzig formalism does not coarse grain by "adding to the equations of motion . . .a deterministic, a stochastic and a non-markovian term". It is a mere reformulation of the dynamics which consists of (not adds) those three terms.

2.) page 3: between lines 5 and 10; the Wasserstein distance (or any distance measuring how different the two pdfs are is not a measure "how different the attractors are". One can construct the same pdf with very different chaotic systems. The authors could maybe replace "attractor" by "statistical behaviour" (although it does not tell you anything about temporal behaviour such as correlation functions).

3.) page 6: R in (20) is not defined/used anywhere.

4) page 7: above (25); \mu and \nu are measures nots sets.

5.) A reader would profit, I believe, from more detailed figure captions. Which system is used (L63 is fast or slow)? What are the parameters?

---

## Referee Comment (RC2) · Anonymous Referee #2 · 9 Apr 2018

This paper explores a scheme for parameterization and tests it on an idealized case. The paper is interesting and the results are convincing, albeit on the test case the authors chose. A few points remain unclear in the rationale of the paper and its formulation.

Major points

The introduction presents the motivation of the paper, which is subgrid parameterizations due to the wide range of scales that occur in the modelling of geophysical fluids. The introduction focuses on climate models, for which parameterizations essentially involve convection and clouds, i.e. a range of spatial scales. But the authors focus

on a temporal diversity of scales. I hence have the feeling that they treat a different problem than the one that is addressed by the works of Palmer et al. on stochastic parameterizations (of physical processes).

Can the authors provide an illustration (or a discussion) of how their results can be adapted to the problems of subgridscale parameterizations? (and not "just" the question of temporal scales).

The authors do not specify how they integrate the system Eqs. (7-12). I guess they use a Runge-Kutta scheme. But given the fact that two time scales are active, they must use an integration time step that is adapted to the fastest one. This problem occurs when coupling ocean and oceanic models, which bear different CFL conditions. When they integrate Eq. (7-9) alone and add a noise, they might chose to use a different time increment. This would be the rationale for parameterization. What is the new time increment? Please give more details on the experimental settings.

Minor points

p. 2, l. 27: The paper does not seem to deal with subgrid phenomena.

p. 2, l. 34: this paragraph states how you plan to solve a scientific question, but you do not mention the precise scientific question you want to address. The scientific question does not seem to that of subgrid scale parameterization.

p. 4, Eqs. (4-5): using lowercase for the Lorenz 63 system makes the reading confusing. The use of upper and lower case for symbols in mathematical works is generally well defined. I suggest using only uppercase.

Section 3: what are $\Psi$, $D$, $S$ and $M$?

l. 18 says that they "indicate" [. . .], but this is not a definition.

The link between $\Psi_K$ and $x$ (from the Lorenz 63 system) has to be guessed to understand Eq. (18). Please make things more explicit.

p. 6, l. 5: what ergodic measure?

Eq. (19): is $\sigma$ related to the $\sigma$ in the Lorenz 63 system? It is not a standard deviation either. What is it?

Eq. (20): what is $\rho$? Why should the average of $\sigma$ be 0?

I do not understand where Eq. (21) comes from. I do not understand why $h$ is always 0.

Table 1: You use $\sigma$ again, but it obviously means something different!

---

## Editor Comment (EC1) · JM Lopez (Editor) · 28 Apr 2018

Dear authors,

you uploaded responses to referee comments and questions that announce modifications in the new paper version, but you have not uploaded the paper files. Please, include a new version of the paper were the changes made are marked clearly so that we can proceed.

---

## Editor Comment (EC2) · JM Lopez (Editor) · 24 May 2018

While I consider the revised paper has improved a lot in readability and the mathematical details now given are very useful, I also think, and concur with the referee on this, the manuscript can widen its audience by paying further attention to the point 1 made by the referee in his/her first report.

The referee rightly pointed out in his/her first report that there is some mismatch between the introduction, where the authors allude to the problem in atmospheric modelling regarding parametrizing phenomena at small SPATIAL length scales while the actual content of the paper deals with parametrizing processes at short time scales

Interactive
comment

with no spatial resolution involved whatsoever. I agree with the referee that it would make it more appealing to the climatologist a brief discussion on how this method could potentially be helpful or extended to the actual problem that climate modellers try to solve, i.e., the parametrization of short spatial scales phenomena that drive long scale physics. I wonder if the authors may foresee an extension of their method to this question and could include some discussion on this. Or, on the contrary, why this is irrelevant, if that is the case.

I would also suggest the authors to keep the original title of the paper, mentioning the Wasserstein distance in the title is perhaps too technical for a title (?). This is of course optional and up to the authors to reconsider.

---

## Author Response (AR1)

The authors are concerned with a system where an L84 system is weakly coupled with a L63 system, applying the recently introduced Wouters-Lucarini (WL) linear response approach to devise a reduced model for the L84-variables.

The authors show numerically that the reduced system is able to reproduce the mean and covariance of the resolved L84 variables of the full system. Furthermore they use the Wasserstein distance to assess the relative merit of a first order and a second order WL parametrization. They show that the WL parametrization provides a reliable parametrization in the case when the driving L63 is a fast driver and when it is slow driver. In particular the latter case is not amenable to classical homogenization theory.

I recommend publication subject to some minor issues:

1.) page 2: The Mori-Zwanzig formalism does not coarse grain by "adding to the equations of motion . . .a deterministic, a stochastic and a non-markovian term". It is a mere reformulation of the dynamics which consists of (not adds) those three terms.

2.) page 3: between lines 5 and 10; the Wasserstein distance (or any distance measuring how different the two pdfs are is not a measure "how different the attractors are". One can construct the same pdf with very different chaotic systems. The authors could maybe replace "attractor" by "statistical behaviour" (although it does not tell you anything about temporal behaviour such as correlation functions).

3.) page 6: R in (20) is not defined/used anywhere.

4) page 7: above (25); \mu and \nu are measures nots sets.

5.) A reader would profit, I believe, from more detailed figure captions. Which system is used (L63 is fast or slow)? What are the parameters?

Nonlin. Processes Geophys. Discuss.,
https://doi.org/10.5194/npg-2018-16-RC2, 2018

[Figure]

This paper explores a scheme for parameterization and tests it on an idealized case. The paper is interesting and the results are convincing, albeit on the test case the authors chose. A few points remain unclear in the rationale of the paper and its formulation.

Major points

The introduction presents the motivation of the paper, which is subgrid parameterizations due to the wide range of scales that occur in the modelling of geophysical fluids. The introduction focuses on climate models, for which parameterizations essentially involve convection and clouds, i.e. a range of spatial scales. But the authors focus

on a temporal diversity of scales. I hence have the feeling that they treat a different problem than the one that is addressed by the works of Palmer et al. on stochastic parameterizations (of physical processes).

Can the authors provide an illustration (or a discussion) of how their results can be adapted to the problems of subgridscale parameterizations? (and not "just" the question of temporal scales).

The authors do not specify how they integrate the system Eqs. (7-12). I guess they use a Runge-Kutta scheme. But given the fact that two time scales are active, they must use an integration time step that is adapted to the fastest one. This problem occurs when coupling ocean and oceanic models, which bear different CFL conditions. When they integrate Eq. (7-9) alone and add a noise, they might chose to use a different time increment. This would be the rationale for parameterization. What is the new time increment? Please give more details on the experimental settings.

Minor points

p. 2, l. 27: The paper does not seem to deal with subgrid phenomena.

p. 2, l. 34: this paragraph states how you plan to solve a scientific question, but you do not mention the precise scientific question you want to address. The scientific question does not seem to that of subgrid scale parameterization.

p. 4, Eqs. (4-5): using lowercase for the Lorenz 63 system makes the reading confusing. The use of upper and lower case for symbols in mathematical works is generally well defined. I suggest using only uppercase.

Section 3: what are $\Psi$, $D$, $S$ and $M$?

l. 18 says that they "indicate" [. . .], but this is not a definition.

The link between $\Psi_K$ and $x$ (from the Lorenz 63 system) has to be guessed to understand Eq. (18). Please make things more explicit.

p. 6, l. 5: what ergodic measure?

Eq. (19): is $\sigma$ related to the $\sigma$ in the Lorenz 63 system? It is not a standard deviation either. What is it?

Eq. (20): what is $\rho$? Why should the average of $\sigma$ be 0?

I do not understand where Eq. (21) comes from. I do not understand why $h$ is always 0.

Table 1: You use $\sigma$ again, but it obviously means something different!
* * *
Nonlin. Processes Geophys. Discuss.,
https://doi.org/10.5194/npg-2018-16-AC1, 2018

[Figure]

We wish to thank the reviewer for the positive review of our paper and for his/her useful comments.

1-2) We applied the modifications requested.

3) We defined $R(t)$ above the corresponding equation.

4) The sentence has been modified accordingly.

5) We added to all the captions the value of $\tau$, pointing out the case of fast/slow Lorenz 63. We think that further additions would be redundant, since all the other model

parameters used throughout the paper remain unchanged.

Nonlin. Processes Geophys. Discuss.,
https://doi.org/10.5194/npg-2018-16-AC2, 2018

[Figure]

We wish to thank the reviewer for his/her useful comments, through which we hope to have improved the paper.

Major points:

1) In this paper we focus on the time scale separation between Lorenz 84 and Lorenz 63 - along with the fact that we can parametrize a slow system with the same approach used for a fast system, thus proving the extreme flexibility of the scheme presented here. In a previous paper (Vissio and Lucarini 2018) we treated more explicitly the case

of parametrizing the effects of the interaction between the large scale, slow variables of interest with the small, fast scales we want to parametrize. Nevertheless, we have modified the introduction to better describe our aim.

2) We have added the scheme and the time increment right below Eqs.7-12. The latter has been set to the standard for Lorenz models (0.005) but, in order to check that this increment was small enough, we have run tests with timestep=0.001. The results found conform very well with what reported in the paper, using the larger time increment 0.005.

Minor points:

1-2) See Major point 1.

3) We have changed the name of Lorenz 63 variables from $x$ to $\widetilde{x}$ (similarly for $y$ and $z$).

4-5) We have specified the meaning of $\Psi$ under Eqs.13-14 and that $D$, $S$ and $M$ are defined by Eqs.18-22.

6) We have written the value of the two couplings at the beginning of Section 3.1.

7) We are referring to the ergodic measure of $\widetilde{x}$ (added in the paper).

8) $\sigma$ is a stochastic noise, we have clarified this right after Eq.19. In order to avoid confusion, we changed the notation to $\omega$.

9) The reviewer is right, we have explained that, e.g., $\rho_{o,x}$ indicate the expectation average with the respect to the measure of the Lorenz 63 system.

10) As written at the beginning of Section 3, the derivation and main results concerning the paramerization scheme is reported in Wouters and Lucarini (2012, 2013, 2016) and further explained in Demaeyer and Vannitsem (2017). Our aim here is just to recap the main findings in order to apply the formulas to the model.

11) $\sigma$ in Table 1 is the standard deviation, as specified in the caption. We have now changed notation in the other two cases (Lorenz 63 parameter, now $s$, and stochastic noise, now $\omega$) and clarified in the text and in the table that the statistics are computed over the ensemble of realisations.

**Changes in manuscript**

- New title, to highlight the importance of Wasserstein distance in our paper.
- Rewrite of several sentences to improve the exposition.
- More detailed figure captions.
- More definitions for parameters and variables used.
- New symbols for parameters with the same name.
- Rewrite of parts of the introduction to expose the focus of our paper on time scale separation instead of subgrid scale.
- More details on time scheme used.
- Partial rewrite of the Lorenz 84 forced by Lorenz 63 subsection.
- Better exposition of Wouters-Lucarini approach, marking as vector the quantities in Eqs.13-22.
- New final sentence.

[revised manuscript text omitted]

---

## Author Response (AR2)

We wish to thank the editor and the reviewers for the positive review of our paper and for the suggestions provided.

In the new version of the paper we have implemented three small modifications:

1) We wish to retain in the title reference to the Wasserstein distance - which is the main point in our paper. We have also added "stochastic" in the title, in order to better chracterize the type of parametrization we are dealing with.

2) Page 3, Lines 6-10: Clearer explanation of the main novelty of the paper.

3) Page 6, Lines 6-12: Discussion about scale adaptivity, with the clarification that we still don't know if the methodology can be applied when dealing with spatial scale separation.

Best Regards,
G. Vissio and V. Lucarini